# Towards Dependable IoT via Interface Selection: Predicting Packet Delivery at the End Node in LoRaWAN Networks

**DOI:** 10.3390/s21082707

**Published:** 2021-04-12

**Authors:** Marc Guerrero, Cristina Cano, Xavier Vilajosana, Pascal Thubert

**Affiliations:** 1Computer Science, Multimedia and Telecommunications Department, Universitat Oberta de Catalunya, 08018 Barcelona, Spain; ccanobs@uoc.edu (C.C.); xvilajosana@uoc.edu (X.V.); 2Cisco Systems France S.à.r.l., 92130 Issy-les-Moulineaux, France; pthubert@cisco.com

**Keywords:** IoT, LoRaWAN, dual-band, dependable, estimation

## Abstract

Estimating channel conditions to predict packet delivery can be exploited as a powerful tool to ensure wireless networks dependability. In this article we explore the practical application of this idea from the end-device perspective, using the LoRaWAN protocol stack. We aim to understand if packet delivery can be estimated considering different levels of feedback at the end-device. For that, an extensive data collection campaign is carried out. Through an analysis of the obtained traces, we establish correlations between connectivity metrics at the end node and the fact that a packet is received at the gateway. The study is complemented considering different levels of feedback: (i) No feedback, (ii) enabling acknowledgements frames, and (iii) considering application/control plane data about the channel status at the gateway side. The results show that it is possible to estimate packet delivery in all the evaluated cases.

## 1. Introduction

The adoption of Low Power Wide Area Networks (LPWAN) is notorious in outdoor large-scale Internet of Things (IoT) application scenarios [1]. LPWANs provide kilometer-range coverage, limited power consumption, and simplified network architecture and management. All these properties come with important restrictions in terms of data-rate and bandwidth, especially in those technologies operating at Industrial Scientific and Medical (ISM) bands. These bands are regulated by regional administrations that impose limitations on their use. For example the ETSI EN300-220 regulation [2] at the 863–870 Mhz band limits the transmission duty-cycle to 0.1% or 1% depending on the channel and medium access policy used.

LoRaWAN [3] is a widely-adopted LPWAN technology. It is an entire protocol stack and architecture that exploits a robust Chirp Spread Spectrum modulation at the physical layer referred to as LoRa. LoRa features a low data rate (in the order of few kilobits per second or even few bits per second depending on the chirp spreading factor and utilized bandwidth). Despite the success of the technology, the community is aware of its limitations [3] with one of the understood weaknesses being performance in dense deployments.

Since LoRaWAN uses an Aloha medium access, collisions increase as the network density increases, limiting its scalability. Moreover, due to duty-cycle regulations, the use of Acknowledgments (ACKs) is usually avoided. These limitations put at risk the reliability of these networks. On the other side, however, the transmission cost in LoRaWAN is usually low compared to other alternatives, such as NB-IoT, in which the subscriber pays a cost per message transmitted.

We argue that exploiting the use of LoRaWAN when the packet delivery rate is high and using alternative technologies when the channel is congested may be beneficial both in terms of cost and dependability of IoT systems. Thus, in this article we explore mechanisms to enable end nodes to smartly decide using the LoRaWAN interface when the channel conditions are favorable. This effort is aligned to the objectives of the Reliable and Available Wireless (RAW) Working Group at the Internet Engineering Task Force [4] in which control plane mechanisms exploiting diversity are designed to improve wireless networks service guarantees. With this goal, we take a learning approach in this work. That is, we are interested in estimating the probability of correct packet reception depending on actual network conditions of the deployment of interest and at the specific time the packet is to be transmitted.

The paper is organized as follows. First, in Section 2 we provide an overview of LoRaWAN, indicating the technological aspects needed to follow the rest of the article. Then, in Section 3 we describe what has been done so far to estimate the channel conditions in LoRaWAN networks. This motivates the need for our study, outlining what directions have not been taken by prior art. We then, in Section 4, discuss our setup and data acquisition process as well as the methodology, identifying the metrics and variables available at the end-devices and at the gateway and the algorithms used for data exploration and prediction. In Section 5, we present the exploratory analysis of the data aiming to identify which features allow the end node to assess the status of the medium. We differentiate cases in which no feedback is available to the end node and cases in which feedback at the link layer or at the application layer are received at the end node. We also analyze in this section the results of different algorithms for predicting whether the packet will be correctly received at the gateway. Finally, some conclusions are given.

## 2. LoRaWAN Overview

Back in 2011, a startup company, Cycleo, developed a robust yet simple communication technology exploiting Chirp Spread Spectrum modulation. This company was acquired by Semtech in 2012. The technology was named Long Range (LoRa) and has become a widely-adopted technology in the Internet of Things (IoT) landscape. In particular, LoRa has been used for outdoor and industrial scenarios requiring long-range coverage. On top of the LoRa modulation, a complete protocol stack was developed. The stack, named LoRaWAN, exploited Aloha access combined with hub-and-spoke network topologies in which gateways can simultaneously receive packets in a pseudo-connection-less fashion. Lately the LoRa modulation has been extended with a new modulation technique exploiting Frequency Hopping Spread Spectrum (FHSS). This new modulation has been referred to as Long Range-Frequency Hopping Spread Spectrum (LR-FHSS) [5]. LR-FHSS can be used interchangeably with LoRa as part of the LoRaWAN protocol stack.

LoRaWAN defines a complete network architecture in which end devices, typically sensor-equipped nodes, communicate with one or more gateway devices. The gateway device acts as a simple layer 2 packet forwarder and relays the information to a back-end server architecture that handles the data packets, forwarding them to proper application. The network server is also in charge of the control plane of the LoRaWAN network, trying to optimize the network operation through adaptive data rate policies and taking care of the network security and authentication procedures.

The LoRa physical layer is engineered to guarantee low power operation and high levels of robustness and coverage (in the order of kilometers) at the cost of limited bandwidth. The data rate is limited, ranging from 22 kbps with spreading factor 7 and a 500-kHz channel width to 292 bps with spreading factor 12 and a 125-kHz channel width, among the most restrictive. As LoRa is based on Chirp Spread Spectrum (CSS) modulation, different chirp redundancy configurations are possible. Those are referred to as Spreading Factors (SF). These different configurations of the physical layer enable to trade off bandwidth utilization to robustness.

LoRaWAN defines different bandwidth configurations per channel (from 125 kHz to 500 kHz) depending on the regional parameters regulations. The Aloha access combined with the regional duty cycle regulations impose constrains to the time on air utilizable per device, considerably limiting the maximum achievable throughput [3]. LoRaWAN defines three different network operation modes, referred to as Classes. In Class A, nodes are power duty cycled and can only receive downlink messages after an uplink window. In Class B, a periodic beacon is sent by the gateway, enabling nodes to loosely synchronize to the beacon and their transmissions allocated in an uplink window. Class C assumes end nodes are powered and listening when not transmitting. One key characteristic of LoRaWAN is that gateway devices can receive packets simultaneously from different channels and using different spreading factors. SFs are considered orthogonal, not interfering from one to the other. This enables the coexistence of different transmitters with collisions only occurring when transmitters coincide in time, frequency, and spreading factor. Even in such a case and thanks to the robust CSS modulation, the LoRa receiver can decode a packet if the signal strength is well below the noise floor.

In LoRaWAN, packets are typically not acknowledged as the gateway downlink capacity is limited by the duty cycle regulations, preventing to acknowledge all possible uplink packets when networks are dense. Yet the LoRaWAN protocol stack enables one to request active packet confirmations for particular frames through a bit field in the header. This mechanism is completely managed by the end device. Acknowledgements are received in the downlink windows opened right after an uplink message. This downlink window can also be used by the gateway to send network or data commands to the end device.

The LoRaWAN server is able to adapt the node’s configuration according to different network performance indicators. An Adaptive Data Rate (ADR) policy can be enabled so that the gateway device suggests the SF and channel to use to a particular device based on link quality indicators, e.g., Received Signal Strength (RSSI), Signal-to-Noise Ratio (SNR), etc. The ADR mechanisms monitors uplink messages from each of the end nodes and through a periodic downlink command informs the device about the SF and channel to use. This policy aims to minimize the network occupancy, configuring the devices to use the smallest SF so as to improve the overall network and device performance as well as the power consumption.

## 3. Related Work

We are interested in predicting whether a packet will be correctly received by a LoRaWAN gateway. Therefore, our interest is in inferring the link performance, which encompasses the propagation characteristics of the channel, interference, as well as collision probability. In this section we overview related work on estimating the link performance of LoRaWAN. As we make use of machine learning techniques, we also overview approaches for link quality estimation that utilize machine learning algorithms, similar to our work.

### 3.1. Channel and Link Quality Estimation in LoRaWAN

Channel and link quality estimation in cellular systems, WiFi, mmWave, and the such can be supported by control channels and/or feedback between the base station/access point and the user equipment/station. The data-intensive characteristic of these systems also makes the metrics used for channel/link estimation often available. For instance, in WiFi, an ACK is received per each packet transmitted. In LoRaWAN networks, in contrast, packet transmissions are sporadic and duty-cycle restrictions impose further limitations on the acquisition of metrics to be used for estimation. A typical LoRaWAN end node may send few packets per unit of time and no ACK will be sent back by the gateway in order to reduce the downlink channel occupation of the network. These characteristics of LoRaWAN limit the capabilities of correctly estimating the link quality at the end-node side, as we aim to do in this work.

In fact, works that focus on link quality estimation in LoRAWAN perform this estimation at the gateway side [6,7]. The goal is usually to improve performance via proper resource allocation or parameter configuration. For instance, Zhou et al. [8] propose a Data Rate and Channel Control (DRCC) scheme for LoRa. The scheme estimates channel conditions based on the short-term Data Extraction Rate (DER), as the ratio between successfully received messages by the LoRa gateway and transmitted messages by the LoRa node within an estimation window [9]. The DER information is calculated at the gateway as a ratio between the number of packets received and the total number of packets sent inferred from the packet counter (FCnt field) of the LoRAWAN header. A short-term window is defined in order to capture the recent status of the channel only. The paper proposes to opportunistically adjust the spreading factor to adapt to the variation of channel conditions. Authors propose a load-balancing strategy based on the knowledge of the current spreading factor and channel allocation. The strategy allocates nodes to the available channels and spreading factors by equalizing the channel and spreading factor load (e.g., based on total time on air).

Another example of channel estimation at the gateway side, in this case path-loss, is the work of Reynders et al. [10], which proposes a method to address the hidden terminal effect caused by transmitters near the gateway to those transmitters that are far from it. The main goal is also the allocation of resources (SF) and optimization of transmission parameters (power). The method aims to minimize the collision rate within the same spreading factor by means of clustering the devices in groups, considering the distance to the gateway based on the estimated path-loss [11]. Path-loss is estimated at the gateway by subtracting the received power to the transmitter power assigned by the base station. The allocation of SF, transmission power, and channel is done in a per-cluster basis, minimizing the chances of collision by allocating different SFs and transmission power to the nodes in the cluster.

A related line of research is the evaluation of the Channel Activity Detection (CAD) functionality provided by the LoRa transducer. The CAD mode efficiently detects the LoRa preamble signal [12] enabling a listener to match its configuration to receive a given packet or avoid a transmission using that configuration. For instance, Kim et al. [13] define an adaptive spreading factor selection between two single-channel LoRa modems. The approach is designed to support efficient multihop operation using low-cost LoRa modems and mainly allocates pairs of transmitters to spreading factors in order to avoid collisions along the multihop topology. The defined policy relies on estimating the spreading factor used by a transmitting node based on the CAD functionality.

Similarly, Kennedy et al. [14] evaluate the use of Clear Channel Assessment (CCA) in LoRaWAN networks. Authors experimentally evaluate the suitability of using CAD as a CCA technique. The study demonstrates that eight consecutive CAD measurements enable the detection of both preambles and LoRa frames at distances up to 4 km. They also show that applying CAD to the preamble brings a 1-dB to 2-dB SNR advantage when compared to its application to the payload. Yet, further work is necessary to show that CAD can be used to predict link quality over long distances.

A recent approach to link quality estimation in LoRa is presented in [15]. Authors propose classifying the environment that traverses a link prior to deployment. The mechanism is thus not adaptive to changing environmental conditions and does not take into account interference with other networks or losses due to collisions with other LoRaWAN devices.

Note that the techniques overviewed above along with the subsequent configuration of the network by the gateway can be used in combination with our approach. LoRaWAN may be optimized from the gateway perspective but it may still be congested, since there is no guarantee on delivery rate. Our approach aims to enable the end node to smartly decide whether to use the LoRaWAN interface at all or use other means for transmission, while the techniques presented assume the LoRaWAN interface is to be used and optimize the network to improve, but not guarantee, delivery.

### 3.2. Machine Learning Techniques for Link Quality Estimation

The use of machine learning techniques to estimate the link quality of a wireless link has received considerable attention by the research community. We refer the reader to Cerar et al. [16] for an exhaustive and up-to-date survey on the topic. According to [16], existing works on link quality estimation using machine learning mainly use either classification or regression techniques. Models such as naive Bayes, logistic regression, Artificial Neural Networks (ANN), e.g., [17,18,19], and more recently, Support Vector Machines (SVM), such as in [20], are commonly used.

Previous work, however, exploit the fact that immediate feedback about the transmission over the link is received by the node performing the estimation, either in the form of SNR, RSSI, or the Link Quality Indicator (LQI) of a packet or ACK. The Packet Delivery Rate (PDR) is also usually immediately available due to the transmission of an ACK per each packet received. These metrics, as we can see in the works overviewed in [16], are the usual inputs of machine learning algorithms used for link quality estimation in wireless networks. In this work, we aim to extend related work to enable an end node in LoRaWAN to estimate whether a packet will be received by the gateway, with limited information: No feedback (only the information that is locally available to the node) or with sporadic feedback (at the link or application layer) sent from the gateway to the end node. So, similarly to previous work, our interests are to perform an exploratory analysis of the data and to predict whether a packet will be received based on available features. However, in our case, the inputs are limited to what can be made available to the end node, taking into account the accuracy-cost trade-off of sending feedback.

## 4. Data Gathering and Methodology

In order to study and evaluate the ability to predict the probability of correct reception by the end node in LoRaWAN, we developed an experimental setup to obtain realistic data, avoiding to rely on propagation models or simulation frameworks. In this section, we first describe the experimental system setup and the performance results of the data set obtained. We then describe the methodology, identifying the metrics and labels to be used in our analysis, as well as the algorithms selected.

### 4.1. Experimental Setup

We considered transmissions from a router impaired by interferer devices nearby. Router and interferer devices were integrated using a Raspberry PI 3b+ single board PC and a Dragino LoRa Extension Hat (see Figure 1). The Dragino Extension Hat communicates through the SPI interface with the Raspberry PI. The setup is mains-powered and connected through an ethernet connection to a local area network in which control and operation tools can be used to manage devices.

A Python-based and software controller has been designed to relay generated traffic to the LoRa radio. The LoRaWAN protocol stack has been fully implemented in Python and instrumented to facilitate the capture of connectivity metrics and traces (The developed tools are made available in https://github.com/wine-uoc (accessed on 10 April 2021)). The LoRa server was deployed in our premises using the Chirp Stack implementation [21]. A set of connectors were developed to extract operational information from both the router devices and server. The information was logged into a database for posterior analysis.

Figure 2 presents a schematic representation of our deployment. A LoRaWAN gateway (Kerlink Wirnet station) is deployed inside a building while the nodes are outdoors. The experimental setup is placed in a residential area where interferences coming from other sources are not significant. Additionally, to reach the gateway, the signal transmitted by the nodes has to go through a brick wall and travel by an indoor path. This has an impact in the path-loss of the channel.

Router and interferer devices generate traffic according to a Poisson distribution. The packet generation rate (λ) is such that with the router and the 4 interferer nodes (number of nodes, n=5) the packet delivery ratio, considering Aloha channel access, results in $PDR={\left(1-\lambda X\right)}^{2(n-1)}\approx 0.4$, with *X* as the frame duration. This results in λ=0.25 packets/s for SF=7 (X=369 ms, PDR=46.1%) and λ=0.035 packets/s for SF=12 (X=2.7 s, PDR=45.2%). The packet generation rate is kept constant for all configurations of the same spreading factor. In addition, we enabled acknowledgement frames for the LoRa uplink packets. This is explicitly done using the confirmable field of the LoRaWAN header. We use two different spreading factors (SF=7 and SF=12) and transmission power configurations ($P_{\mathrm{tx}}=0$ dBm and $P_{\mathrm{tx}}=14$ dBm). We vary the number of interferers ($n_{\mathrm{int}}$) from 1 to 4 for each experiment. We consider homogeneous conditions (all nodes, router and interferers, use the same configuration parameters).

#### 4.1.1. Performance Results

Figure 3 shows the PDR of our setup when $n_{\mathrm{int}}$ varies from 0 (only the router is transmitting) to 4, effectively increasing the collision probability, for the values of SF and $P_{\mathrm{tx}}$ considered. The figure depicts the PDR obtained by each node in the uplink, as well as the average of the PDRs of all nodes and the theoretical value. First, we can see that experimental and theoretical results are very close, validating our experimental setup. We also observe that increasing power from 0 to 14 dBm for the same SF (i.e., comparing Figure 3a with Figure 3b and Figure 3c with Figure 3d) does not result in an increase of PDR. The reason for this is that, as we consider homogeneous conditions, packets are received at a higher power at the gateway but so do colliding packets. Additionally, the channel conditions in the experiments are good due to the proximity of the devices. As a result, considering non-homogenous conditions (although not shown here) provide similar results.

We can also observe that for SF=12, the average PDR obtained becomes considerably higher (5% on average) than the theoretical result as the number of interferers increase when $P_{\mathrm{tx}=0}$ dBm (see Figure 3c). However, this does not occur for $P_{\mathrm{tx}}=14$ dBm. Our hypothesis is that the capture effect is occurring and that it allows more packets to be decoded in case of collisions for SF 12 due to its robustness. However, the capture effect may be more limited when the power of the colliding packets increases. We checked the magnitude of the capture effect for scenarios with higher collision probability, showing results compatible with this hypothesis.

Since all packets are acknowledged we can observe the downlink performance. Figure 4 shows the percentage of ACKs received per packet transmitted in the setup and experiments shown in Figure 3. If we compare both figures, which gives us insight on the performance of the uplink (Figure 3) and downlink (Figure 4), we can observe that for SF=7 the PDR of the downlink is reduced if compared to the PDR of the uplink. The same does not occur for SF=12, for which the performance of the downlink is similar to that in the uplink. This behavior can be explained by a certain degree of asymmetry between the transceiver characteristics at both ends. The radio chipset used in the gateway devices (e.g., sx1301) features higher sensitivity than transducers used at the end nodes (e.g., sx1276). As a consequence, this is making the end node less able to successfully decode packets transmitted using a weaker SF. This fact must be accounted for in our subsequent analysis as considering a successful reception will depend on the device’s point of view.

### 4.2. Methodology

We detail here the methodology we have used in this work, in terms of the different metrics considered as input and labels to our algorithms as well as the algorithms considered in our subsequent analysis and evaluation.

#### 4.2.1. Input Data and Labels

In our exploratory and prediction analysis we consider the following data for each transmitted packet (When a metric is not available, we set it to −999). Note that the metrics taken per packet are those that are available before the packet transmission. These metrics are directly obtained from the radio chipset at the end-node without requiring any other tool. The gateway information is obtained through the LoRa Server operational information.

Spreading Factor (SF): SF that is used to transmit and receive the data.Transmission power ($P_{\mathrm{tx}}$): Power used to transmit the packet.Noise Floor (*N*): RSSI sample from the channel immediately before transmission. We also consider the sample taken for the last packet transmitted ($N_{l}$).Indicator of packet received ($I_{o}$): boolean variable that indicates whether a packet from nodes other than the gateway is received during the CCA check previous to transmission. We also consider this variable for the last packet transmitted ($I_{o,l}$).SNR of packet received (${\mathrm{SNR}}_{o}$): SNR from the packet received during the CCA check previous to transmission (if any). We also consider the sample taken for the last packet transmitted (${\mathrm{SNR}}_{o,l}$).RSSI of packet received (${\mathrm{RSSI}}_{o}$): RSSI from the packet received during the CCA check previous to transmission (if any). We also consider the sample taken for the last packet transmitted (${\mathrm{RSSI}}_{o,l}$).Signal-to-Noise ratio (${\mathrm{SNR}}_{\mathrm{ack},l}$): SNR from the acknowledgment received from the gateway (if the acknowledgement has been received) for the last packet transmitted.Received Signal Strength Indicator (${\mathrm{RSSI}}_{\mathrm{ack},l}$): Signal strength from the acknowledgment received from the gateway (if the acknowledgement has been received) for the last packet transmitted.SNR of packet received at the gateway (${\mathrm{SNR}}_{\mathrm{gw}}$): SNR from the last packet received at the gateway (if the packet has been correctly received).RSSI of packet received at the gateway (${\mathrm{RSSI}}_{\mathrm{gw},l}$): RSSI from the last packet received at the gateway (if the packet has been correctly received).

As for label variables, we have two options:Indicator of packet received at the gateway ($I_{\mathrm{gw}}$): Boolean variable that indicates whether the packet has been correctly received at the gateway.Indicator of ACK received ($I_{\mathrm{ack}}$): Bboolean variable that indicates whether an ACK is received from the gateway acknowledging the correct reception of this packet (if feedback at the link/application level is used).

As can be seen, part of the information is already available at the end node without requiring any signaling from the gateway device. However there are variables that can only be obtained if the network implements acknowledgement frames or if it exists an application/control plane mechanism to explicitly inform about gateway-side performance metrics. We categorize this cases as follows: (i) No feedback (when metrics related to the ACK or measured at the gateway are not considered/not available), (ii) link-level feedback (when metrics related to the ACK are considered or available at the end node), and (iii) application-level feedback (when metrics that are measured at the gateway are made available to the end node). Thus, we are able to differentiate among the following cases:Case A: Without feedback with label $I_{\mathrm{gw}}$.Case B: With feedback at the link level with label $I_{\mathrm{gw}}$.Case C: With feedback at the application level with label $I_{\mathrm{gw}}$.Case D: Without feedback with label $I_{\mathrm{ack}}$.Case E: With feedback at the link level with label $I_{\mathrm{ack}}$.Case F: With feedback at the application level with label $I_{\mathrm{ack}}$.

These cases allow us to differentiate among conditions of no feedback and feedback at different levels and evaluate accordingly the improvements on accuracy and cost.

As our goal is to estimate the channel status from an end-device perspective and LoRaWAN is mainly an uplink technology, we consider the case when there is no feedback after a transmission of a packet. In this sense, the channel estimation should be performed only with the knowledge related to the current configuration and what can be perceived by the node through CCA checks and link status metrics obtained from the radio. Recall that CCA turns on the radio to listen before any transmission. If the noise level perceived in this listening window is above a threshold, the node considers the channel is occupied by another transmitter. In the scenario considered, nodes are separated by relatively short distances compared to what LoRaWAN networks allow. However, we consider that similar results may be obtained in networks where nodes are far apart but use the CAD technique as shown in [14]. For this case we use the metrics defined in Table 1 (Cases A and D). As a side note, we want to point out that performing a CCA has an impact in the energy consumption of the node. Following the results obtained in [14], an eight symbol-duration CCA may lead to a 17% increased radio active time in the worst case and affect proportionally to the overall energy consumption of the device.

We then evaluate the results when feedback at different levels are available. If we consider that end nodes receive an ACK per packet transmitted, we can then make use of metrics related to the ACK reception, as shown in Table 1 (Cases B and E). If we consider that application-level metrics can be transmitted to the end node, we can then make use of metrics measured at the gateway for the last received message, see Table 1 (Cases C and F).

#### 4.2.2. Algorithms Description

Our aim is to determine if a binary variable (i.e., the packet/ACK will be received at the gateway/end node or not) can be predicted considering the set of input variables described in the previous section. This can be seen as a classification problem for which numerous techniques exist.

Well-known classification approaches are based on supervised methods that once trained are able to classify inputs into the desired categories. In addition, unsupervised methods are commonly used to gain knowlegde in the available data. In this work we have evaluated relevant methods and combinations to validate our initial hypothesis.

First, the Principal Component Analysis (PCA) [22] algorithm is used to perform a dimensionality reduction. The goal of PCA is to project the data into a lower-dimensional space. The maximum variance is searched when the new dimensional axes are calculated. By doing this, the largest amount of information is kept. PCA can also be used to get insight into the data, since the maximum variance is seeked, a feature contribution can be determined by the weight on each axis.

The second technique considered exploits the combination of a supervised Linear Discriminant Analysis (LDA) [23] classifier and an unsupervised clustering method (K-Means) in order to support packet delivery estimation given a set of known metrics at the node side. LDA is a widely-used technique to reduce the data dimensionality maximizing the separability of the known classes. The procedure is based on maximizing the distance between the means of each category while minimizing its scatter. K-Means [24], on the other side, is one of the most well-known algorithms for unsupervised clustering. The objective is to assign data samples to clusters. The goal of the assignation is to make groups of equal variance, minimizing the within-cluster sum-of-squares criteria, known as inertia. In this method the number of clusters must be specified in advance. This implies the need of having some insights on the data before the execution. Many techniques are used to choose the right number of clusters, such as the elbow rule [25].

Another remarkable method for classification uses Support Vector Machines (SVM) [26] classification algorithm. SVM relies on the Maximal Margin Classifier which seeks for an hyperplane which separates the classes and maximizes the distance to the closest training instances. The hyperplanes that stand closer to the sample of the different classes are called support vectors. To be able to perform well in nonlinearly-separable datasets, a feature mapping to a higher-dimensional space is performed. The idea is to be able to linearly separate the data on this new space. This is commonly known as a kernel trick and it is one of the most powerful tools of this method.

Finally, for the type of addressed problem, a Random Forest classifier [27] becomes an adequate tool, since it relies on a decision tree classifier. Those methods are easy to understand and very intuitive, being able to also give insights on the data. The idea of a decision tree is to recursively use features to evaluate a certain sample of data to be classified. Random Forest trains a large number of decision trees using different parts of the training set. The objective is operate as an ensemble, taking every decision tree output and getting the most repetitive one as the final output. As a final remark, it is important to mention that these methods are computationally intensive for large data sets and cannot be embedded in small computing devices in their offline version. Their online versions however can be incrementally trained in IoT devices, enabling its adaptation based on small sample windows, thus limiting the computational and memory requirements overhead.

## 5. Exploratory Analysis and Data Prediction

In this section we present an exploratory analysis of the data with the goal to analyze its separability according to whether packets are received/acknowledged. We also analyze the results of packet delivery prediction from the end node perspective considering the cases introduced in the previous section.

### 5.1. Exploratory Analysis

In order to infer whether a frame will be received at the gateway considering the aforementioned cases it becomes crucial to understand the correlation between the available variables at the end node and the labels (whether a packet is received at the gateway/acknowledged). This will help us to determine under which conditions it is better to operate, what sort of protocol and control plane signaling would be required, and at which cost, both in terms of network usage (duty-cycle, protocol overhead) and the accuracy of results in terms of packet delivery prediction. For that we present in a descriptive manner the analysis of the obtained variables during our extensive evaluation campaign. We have considered well-known algorithms, presented in Section 4: PCA, LDA+K-Means, and Random Forest.

Figure 5 show the results of applying PCA to our data for Case A and Case D. Only results for these cases are shown because we have obtained very similar results for all cases. The figure shows a representation of the data on the first and second Principal Component (PC) of the six PCs the algorithm has been asked to find. As can be seen in the figure, data is not clearly separable and a classification algorithm (not shown here) performs poorly on this data. Figure 6 shows the explained variance of each PC. As can be observed, for all cases considered, PC1 retains the most explained variance.

Figure 7 presents the results of applying a clustering algorithm like K-Means on the output of an LDA classifier on the dataset with the goal to improve the results obtained by PCA. Recall that the LDA classifier finds the linear combination of variables such that the separation of the classes under study is maximized. In our case those are the correct reception at the gateway side and the fact the ACK is received at the end node, depending on the label considered. K-Means then groups samples on this variable combinations according to their similitude. As for PCA, similar results are obtained for all cases, so we have opted to show only results for Case A and Case D in the figure. We can observe that different clusters become now clearly distinguishable. Table 2 presents the proportion of samples on each cluster (D) as well as the packet delivery probability on each of them (Lost, Recv) for all cases. These results lead us to infer that categorization of the input metrics can be mapped with a certain probability to the desired outcome, i.e., estimating the packet delivery. This will be further discussed in the next subsection.

Aiming to gain more insight into the data, we also applied the Random Forest algorithm. Figure 8 presents for each of the evaluated cases the weight given by Random Forest to the different metrics. As can be observed, for Cases A, B, and C, in which we use the reception of the frame at the gateway as a label, the most discriminant variable is the noise floor sample taken before transmission during CCA. When we consider whether an ACK has been received for that packet as a label, we can observe that the noise floor sample taken during CCA is still important but now the fact that a packet is received during CCA plays a more important role. Receiving a packet during CCA is an indicator of higher activity on the channel. Recall from Section 4 that we effectively considered different traffic loads (the packet generation rate was kept constant while varying the number of nodes). We also show that collisions were affecting more in the transmissions in the downlink. Thus, it is to be expected that when our ground truth is the reception of an ACK, detecting an ongoing transmission during CCA will play a more important role for downlink transmissions. We can also observe that the spreading factor plays now a role, recall that in Section 4 we observed different behavior on the percentage of ACKs received per packet transmitted for the two SFs considered.

It is important to note in Figure 8 that the metrics measured at the node have a bigger impact than those obtained from the gateway. We believe that this result is due to the fact that the metrics at the end node side are more recent with respect to the current packet transmission than those that can be made available by the gateway, which belong to previously transmitted packets.

In this section we have seen that algorithms such as PCA do not help towards data separability of our data set. However, LDA is able to provide better results towards data separability, as shown by the results of a K-Means algorithm on the output of LDA. The use of Random Forest has allowed us to better understand the data. We have seen that conclusions observed in our performance analysis are captured by Random Forest. We have also observed that metrics measured at the end node are given more importance, presumably because these metrics are more recent with respect to the current transmission than those that can be obtained from the gateway. This has important implications in terms of implementing mechanisms to provide the node with feedback. We will evaluate this aspect in more detail next.

### 5.2. Prediction of Packet Delivery at the End Node

In this section we study the predictability of packet delivery from the end node perspective taking into account the characterization done in the previous section. We explore both the predictability that a gateway receives a packet and the predictability that an end node receives and acknowledgement, as those are our ground truth variables. We consider the same scenarios as in previous analysis (Cases A–F), characterized by the absence of feedback, the presence of feedback related to received acknowledgement packets, or the presence of application-level feedback embedded in downlink packets.

#### 5.2.1. Offline Learning

To estimate the model’s performance, the data obtained in the experiment phase has been split into two subsets. The first subset represents the 70% of the data and it is used to train the models, and the second, which represents the 30%, is used to calculate the accuracy of each model. This technique is widely used to evaluate the model in samples that have not been used in the training step avoiding to use data that the algorithm could have seen before.

In this case, it has to be considered that not all the data follows the same pattern because it has been obtained from different network sizes as described in Section 4.1. The methodology, used to split the data, has consisted of splitting the 30% of previously shuffled samples from each single scenario to the validation data and the other 70% to the training subset. The main reason to do it relies on having the most homogeneous subsets of data, increasing the avoidance of big deviations in the prediction results between runs. By doing this, we are sure that the algorithm has seen every scenario in the training stage.

In this section, the results of the different classification methods are presented and compared in order to provide insight on the potential to predict the packet delivery from an end-node perspective as well as to evaluate the improvements (if any) of enabling feedback at the link/application layer.

In Figure 9, we present a comparison between the different evaluated classification methods overviewed in Section 4. Results in Figure 9 show the average of running each algorithm five times with randomly-selected training and input data from our data set (confidence intervals too small to be shown). We compare the results obtained from each algorithm with a base model, that is, always predicting that the packet will arrive at the gateway/be acknowledged. Accuracy is calculated as the packet delivery rate of the packets that are predicted to be received successfully by the algorithm. Thus, the accuracy of the base model corresponds to the PDR. Despite the performance of all algorithms being considered similar, we can derive several conclusions. First, using a prediction method makes the transmitter node improve its PDR when compared to the base model in all cases evaluated. In particular we can observe that a 15% increase can be obtained in the cases in which the predictor estimates the reception of the packet at the gateway (Cases A–C). The use of link-level (Case B) or application-layer feedback (Case C) slightly increase the estimation accuracy compared to no feedback (Case A), but in general the difference is minor. One possible conclusion for this matter is the fact that the feedback information obtained in the gateway is geographically and temporally less correlated with the output than the information that can be obtained at the end node’s side, especially, when a future packet has to be sent, as we have seen in our exploratory analysis. For the cases in which the end node aims to predict if an acknowledgement packet would be received (Cases D–F), we observe a major gain (64%). This considers the fact that the accuracy of the base level in this case is lower (i.e., the probability to receive an ACK at the end node is smaller than the probability that a gateway receives a packet from the end node). In addition, the accuracy of the prediction of the algorithms is higher than what is obtained predicting the correct reception at the gateway. We argue this can be explained due to the ability of the node to detect high traffic load during CCA and the bigger impact collisions have on the downlink, as seen in our performance evaluation. In addition, we can observe a small gain when considering acknowledgement and application layer feedback (Case E and F) compared to no feedback (Case D) because the aforementioned aspects related to geographical and temporal smaller correlation of the information obtained from the gateway.

In Table 3 and Table 4, F1 score [28], which is a weighted average of the prediction and recall, False Negatives (FN), False Positives (FP), True Negatives (TN), and True Positives (TP) are obtained to get more insights on the performance of each model in this classification task.

Models tend to fail more often at classifying a message when it is not received. This is reflected in false positives errors, which are higher than the false negatives. It can be caused because the dataset is not perfectly balanced, showing more cases of packets arriving. In addition, all models obtain similar punctuation in a F1 score showing a tendency in following the same behavior in misclassification. This metrics confirm what is observed in the accuracy comparison, showing a similar performance in the classification of the samples.

We have seen in this section that improving packet delivery ratio exploiting local information at the end node is possible in the conditions of our experimental setup. While this can be considered promising results there are certain considerations that have to be taken into account. First, each device had to be trained/calibrated with previously obtained communication traces. Unsupervised methods, such as PCA, failed to ease data separability in order to discriminate among successfully/failed packet delivery. Supervised methods, in contrast, have proved more successful at discriminating and predicting the outcome of a transmission, improving the packet delivery ratio of the packets transmitted in all studied cases.

In our initial hypothesis, having information about the channel status at the gateway side seemed a good booster to support decisions at the end node. Our results however, indicate that this is not so relevant. We learned that the information at the end node is relevant enough to discriminate correct reception. Adding in information about the channel status inferred from packets received at the gateway does not result in a significant improvement in prediction at the end node. This is related to the fact that information at the receiver end can only be obtained for packets previously transmitted, which is less correlated than information that can be obtained at the end node right before transmission. This result is influenced by our setup, in which nodes were relatively close. The extension to these results to networks where transmitters are far apart is left as future work, and remains to be seen whether the CAD technique may provide a similar detection capability as the CCA technique.

#### 5.2.2. Online Learning

In this section we aim to investigate if an online learning mechanism would lead to similar results as presented in the previous subsection, where results were obtained offline. To that end, an online learning method that updates the model every time that an end-node transmits a packet, has been used. Recall that online learning is used when data are obtained sequentially. The benefit of applying such methodology is to adapt the model’s predictions when data becomes available. We selected Random Forest as it has shown similar levels of accuracy compared to the rest of algorithms evaluated in Section 5.2.1. Plus, Random Forest shows the lowest training time when compared to the others. Figure 10 shows a schematic overview of the operation of the online approach.

In this experiment, the dataset has been divided according to its LoRa radio configuration (SF and transmission power) in order to get more insight from the results.

Moreover, since the online method needs feedback to fit the model, Case D has been considered to approach a more realistic scenario.Due to this, the $I_{\mathrm{ack}}$ label is used as a target variable. Note that sporadic feedback can also be implemented in order to feed the labels to the algorithm in batches. This has been left to future work. In order to initialize the algorithm, an offline training has been done using 10% of the data before starting the online mode. Also, to obtain statistical significance in the Random Forest initialization, five experiment executions were carried out.

Figure 11 shows the accumulated accuracy when using the Random Forest predicted outcome (i.e., transmission success) and the base model in the online learning approach. The figure presents the results for the different considered configurations of the LoRa radio. It can be seen that at the beginning of the experiment there is more instability until the model is fitted with more data. Once this unstable stage passes, convergence is reached. In addition, when compared to the base model, it is shown that the lower the PDR in the base model, the better the accuracy gain achieved by using the predictions obtained from the Random Forest. This is because the Random Forest model chooses more often not to transmit when the channel is highly unstable. When the channel conditions are changing because of high traffic (n>3), we can observe that base model obtains low accuracy due to the more variable results in packet delivery. In general, Random Forest, once converged, shows accuracies between 60% and 80% in all cases evaluated. It has to be taken into account that this result is obtained by reducing the number of packets transmitted through this interface. This means that using a predictive algorithm allows to save energy by not transmitting packets that would be most probably lost.

In summary, it has been demonstrated in an online setting that after a certain amount of samples and with small offline training (only 10% of the data), accuracies similar to offline training can be obtained. In addition, it has to be considered that this proposal is only proven in case the end-node receives feedback, otherwise, an online training can not be triggered. This could cause deviation in case the downlink channel shows a different robustness compared to the uplink channel, avoiding the proper interface exploitation. Methods to provide the model with sporadic feedback and training in batches could be investigated to implement online prediction in a no-feedback scenario.

## 6. Conclusions and Future Directions

In this article we aimed to infer packet delivery from an end-node perspective in a LoRaWAN experimental network. For that, an extensive data collection campaign was carried out. A subsequent analysis of the obtained traces led to the application of different data classification methods to establish correlations between connectivity metrics at the end node and the fact that a packet was received at the gateway. We complemented the study considering different levels of feedback, i.e., considering that packets could be acknowledged and introducing application/control plane data about the channel status at the gateway side. The proposed classification methods were used as PDR estimators considering the available information for each of the evaluated scenarios. Our most relevant results indicate that a 15% PDR gain could be obtained by only considering the channel status information at the end node side. When channel status information at the gateway side was introduced, the predictability of the PDR slightly increased. The increase however was not relevant compared to the cost of sharing such information with the end node. In the evaluated scenario, this small improvement was attributed to the irrelevance of the gateway information at the moment an end node was transmitted since this information was less correlated to the output than the information right before transmission at the end node. We also showed that the results obtained in the offline evaluation could also be obtained in an online setting, executing the algorithm each time new data was available.

Considering the possible directions to extend this work we foresee opportunities to apply the studied methods to different technologies, for instance WiFi, IEEE 802.15.4, or Bluetooth Low Energy (BLE). These technologies typically provide an Automatic Repeat-reQuest (ARQ) mechanism that acknowledges the frames, and the predicted outcome of the transmission can be matched with whether or not an acknowledgement was received. This can be used to continuously learn and refine the prediction; it is possible that some of the lessons learnt here, e.g., the applicability of radio metrics of specific algorithms, applies to other technologies as well and so can be successfully retrofitted. Considering other technologies, the dataset can also be enhanced with metrics that cannot be captured for LoRaWAN. For instance, MIMO radios provide a sense of angle that can be fed in the model. The presence of beacons and their density can be logged, and the rate adaptation can be recorded as well, including the observation of the modulation and coding schemes (MCS), the guard interval, and the channel width that are being used. Another perspective would be to apply the technique over multiple hops of a same technology, for example, across a Wi-Fi or an IEEE 802.15.4 (6TiSCH) mesh. Even if one hop provides a local ARQ, the system is blind over multiple hops, and the situation becomes similar as LoRAWAN, with no feedback on end-to-end success or failure. This area of work is inline with the directions taken by the IETF RAW standardization group. Finally, another axis could be to analyze the influence of many parallel adaptations, based on predictions, and done over transmissions that are not fully independent, e.g., have a common node or a link on path, or use the same spectrum, and may affect one another. Typical but undesirable effects may be observed, and additional techniques could be devised to dampen those effects.

## Figures and Tables

**Figure 1 sensors-21-02707-f001:**
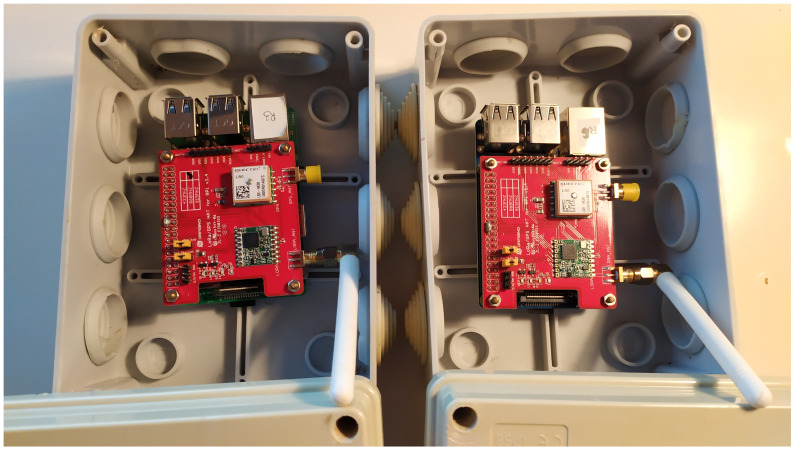
Nodes used during the experimentation campaign.

**Figure 2 sensors-21-02707-f002:**
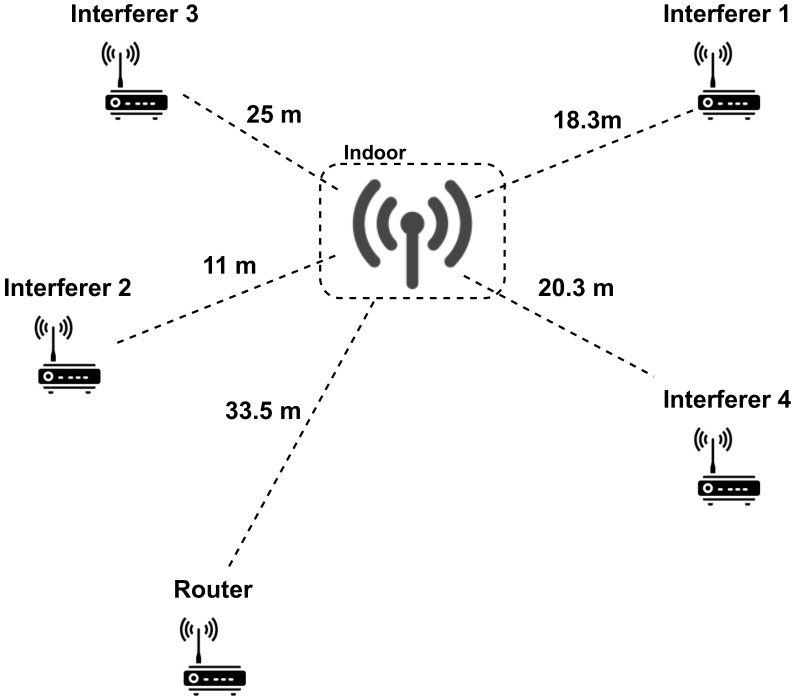
Experimental scenario.

**Figure 3 sensors-21-02707-f003:**
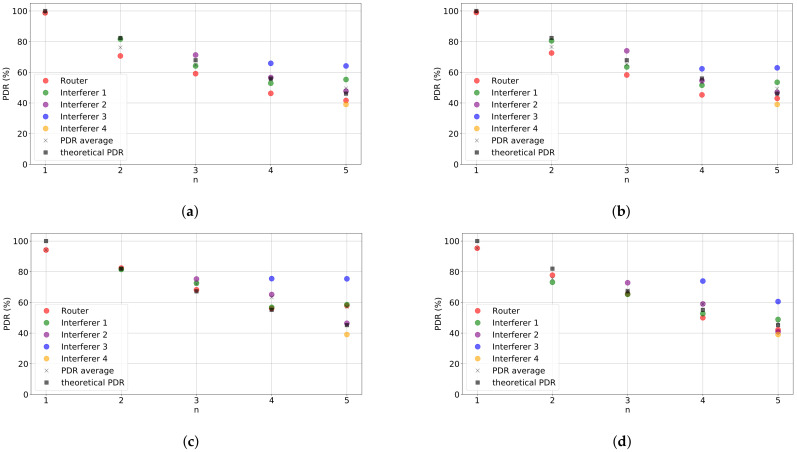
Performance evaluation results. Packet Delivery Rate (PDR) vs number of nodes in the network for different configurations: (**a**) Spreading Factor (SF)=7, $P_{\mathrm{tx}}=0$ dBm, (**b**) SF=7, $P_{\mathrm{tx}}=14$ dBm, (**c**) SF=12, $P_{\mathrm{tx}}=0$ dBm, and (**d**) SF=12, $P_{\mathrm{tx}}=14$ dBm.

**Figure 4 sensors-21-02707-f004:**
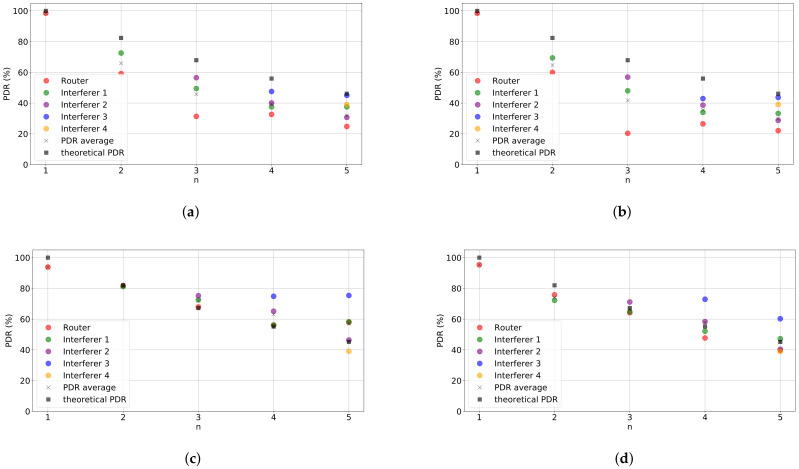
Performance evaluation results. Percentage of Acknowledgments (ACKs) received per packet transmitted vs. number of nodes in the network for different configurations: (**a**) SF = 7, *P*_tx_ = 0 dBm, (**b**) SF = 7, *P*_tx_ = 14 dBm, (**c**) SF = 12, *P*_tx_ = 0 dBm, and (**d**) SF = 12, *P*_tx_ = 14 dBm.

**Figure 5 sensors-21-02707-f005:**
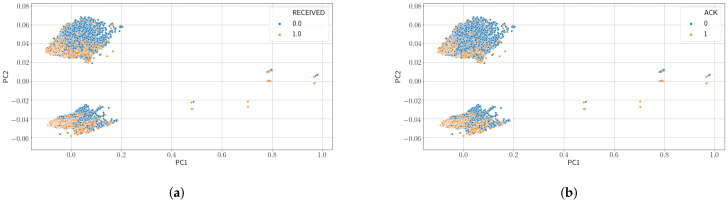
Results from Principal Component Analysis (PCA). (**a**) Using no feedback and *I*_gw_ as label (Case A) and (**b**) using no feedback and *I*_ack_ as label (Case D).

**Figure 6 sensors-21-02707-f006:**
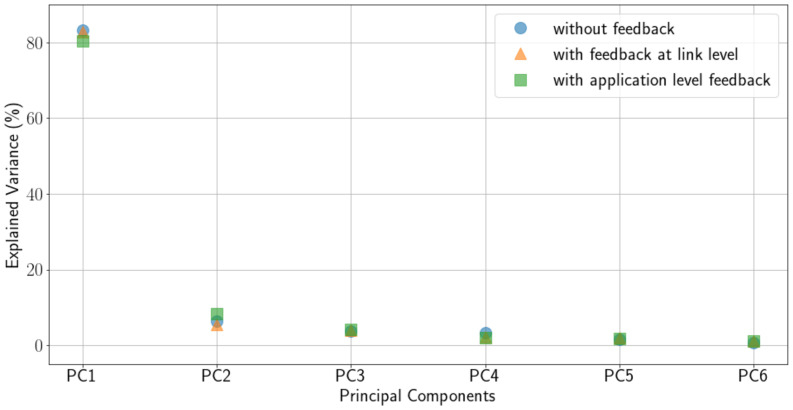
Explained variance of the different principal components in PCA.

**Figure 7 sensors-21-02707-f007:**
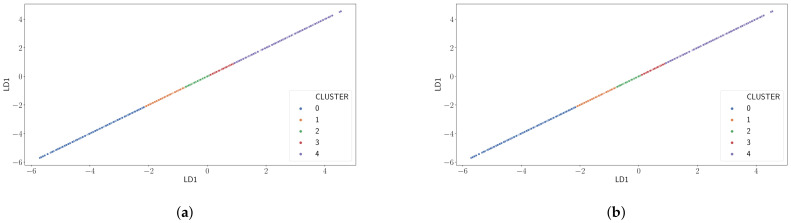
Results from Linear Discriminant Analysis (LDA)+K-Means. (**a**) Using no feedback and *I*_gw_ as label (Case A) and (**b**) using no feedback and *I*_ack_ as label (Case D).

**Figure 8 sensors-21-02707-f008:**
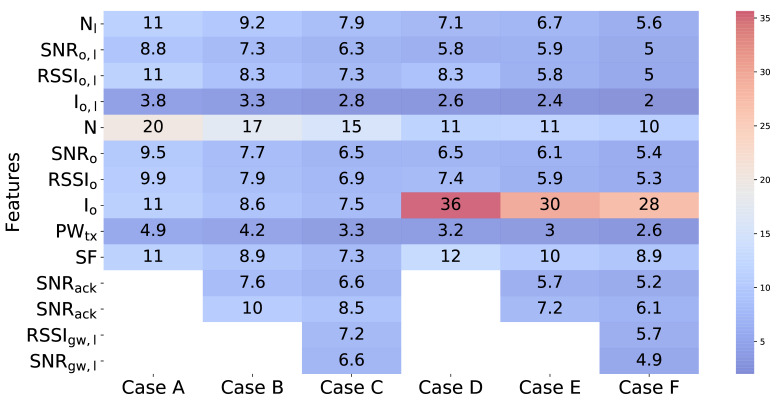
Feature importance obtained from Random Forest for the different cases evaluated.

**Figure 9 sensors-21-02707-f009:**
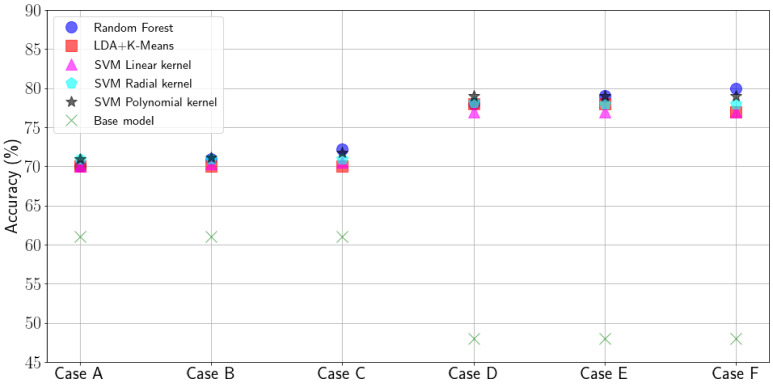
Comparative of the prediction accuracy obtained by the evaluated classifiers.

**Figure 10 sensors-21-02707-f010:**
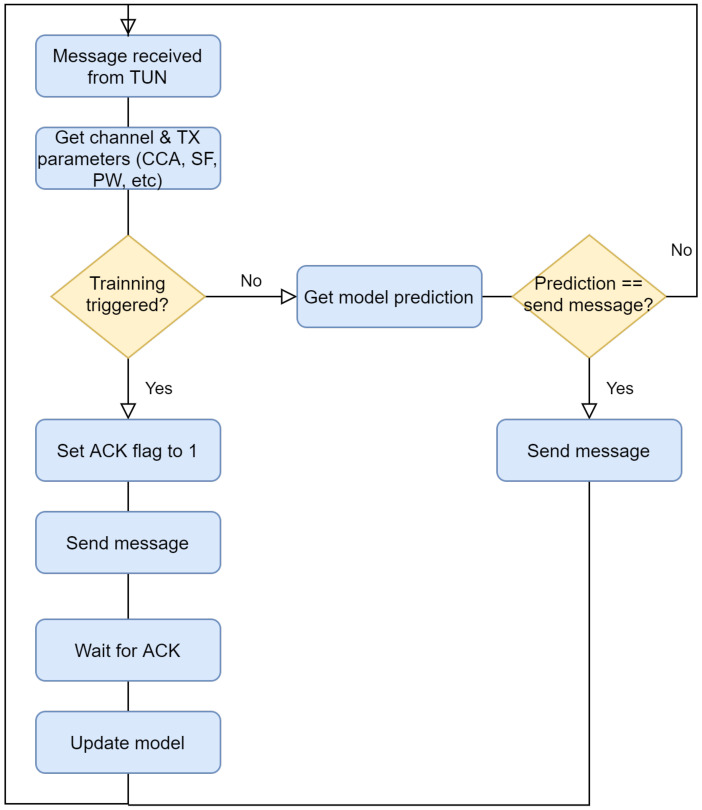
Diagram of the operation of the proposed method.

**Figure 11 sensors-21-02707-f011:**
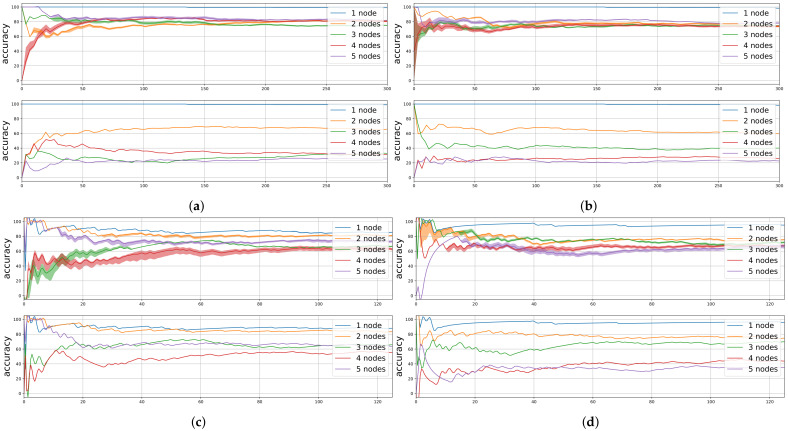
Simulator results: (**a**) SF = 7, *P*_tx_ = 0 dBm, (**b**) SF = 7, *P*_tx_ = 14 dBm, (**c**) SF = 12, *P*_tx_ = 0 dBm, and (**d**) SF = 12, *P*_tx_ = 14 dBm. Top: Random Forest results, bottom: Base model results.

**Table 1 sensors-21-02707-t001:** Metrics used for each case considered. RSSI: Received Signal Strength; SNR: Signal-to-Noise; CCA: Clear Channel Assessment.

Description	Case A and D	Case B and E	Case C and F
Tx power	$P_{\mathrm{tx}}$	$P_{\mathrm{tx}}$	$P_{\mathrm{tx}}$
Spreading factor	SF	SF	SF
Noise floor before tx (last packet)	$N_{l}$	$N_{l}$	$N_{l}$
Indicator of packet rx in CCA (last packet)	$I_{o,l}$	$I_{o,l}$	$I_{o,l}$
SNR packet rx in CCA (last packet)	${\mathrm{SNR}}_{o,l}$	${\mathrm{SNR}}_{o,l}$	${\mathrm{SNR}}_{o,l}$
RSSI packet rx in CCA (last packet)	${\mathrm{RSSI}}_{o,l}$	${\mathrm{RSSI}}_{o,l}$	${\mathrm{RSSI}}_{o,l}$
Noise floor before tx	*N*	*N*	*N*
Indicator of packet rx in CCA	$I_{o}$	$I_{o}$	$I_{o}$
SNR packet rx in CCA	${\mathrm{SNR}}_{o}$	${\mathrm{SNR}}_{o}$	${\mathrm{SNR}}_{o}$
RSSI packet rx in CCA	${\mathrm{RSSI}}_{o}$	${\mathrm{RSSI}}_{o}$	${\mathrm{RSSI}}_{o}$
SNR of the ACK (last packet)		${\mathrm{SNR}}_{\mathrm{ack},l}$	${\mathrm{SNR}}_{\mathrm{ack},l}$
RSSI of the ACK (last packet)		${\mathrm{RSSI}}_{\mathrm{ack},l}$	${\mathrm{RSSI}}_{\mathrm{ack},l}$
SNR of the packet rx at the gateway (last packet)			${\mathrm{SNR}}_{\mathrm{gw},l}$
RSSI of the packet rx at the gateway (last packet)			${\mathrm{RSSI}}_{\mathrm{gw},l}$

**Table 2 sensors-21-02707-t002:** Summary of the clusters distribution for the evaluated cases in LDA+K-Means. Density of data (D), percentage of packets not received/acknowledged (Lost), and percentage of packets received/acknowledged (Recv) in the cluster.

	Cluster 0			Cluster 1			Cluster 2			Cluster 3			Cluster 4		
	D	Lost	Recv	D	Lost	Recv	D	Lost	Recv	D	Lost	Recv	D	Lost	Recv
Case A	4.41	95.53	4.47	17.96	64.96	35.04	23.18	45.31	54.69	34.63	26.55	73.45	19.82	14.32	85.68
Case B	4.62	94.78	5.22	18.21	65.61	34.39	22.88	44.58	55.42	35.19	26.36	73.64	19.11	13.67	86.33
Case C	4.51	94.92	5.08	16.72	66.9	33.1	23.66	45.72	54.28	36.21	26.44	73.56	18.9	13.52	86.48
Case D	3.49	99.06	0.94	21.05	89.93	10.07	11.27	84.37	15.63	37.44	37.76	62.24	26.76	19.78	80.22
Case E	3.38	98.88	1.12	20.79	90.56	9.44	11.91	83.63	16.37	37.44	37.55	62.45	26.48	19.37	80.63
Case F	3.39	98.9	1.1	21.85	90.69	9.31	11.32	83.03	16.97	37.82	36.74	63.26	25.62	18.96	81.04

**Table 3 sensors-21-02707-t003:** Accuracy metrics for base model, LDA + K-Means, and Random Forest. F1 score (F1), False Negatives (FN), False Positives (FP), True Negatives (TN), and True Positives (TP).

	BASE MODEL					LDA + K-MEANS					RANDOM FOREST				
	**F1**	**FN**	**FP**	**TN**	**TP**	**F1**	**FN**	**FP**	**TN**	**TP**	**F1**	**FN**	**FP**	**TN**	**TP**
**CASE A**	47%	0	18,888	0	30,332	71%	7157	7288	11,599	23,176	70%	5954	8513	10,374	24,379
**CASE B**	47%	0	18,888	0	30,332	71%	7568	6941	11,946	22,765	71%	5653	8439	10,448	24,680
**CASE C**	47%	0	18,888	0	30,332	71%	7345	7056	11,831	22,988	72%	4837	8554	10,333	25,496
**CASE D**	32%	0	25,259	0	23,961	79%	2163	8285	16,973	21,799	78%	4179	6492	18,766	19,783
**CASE E**	32%	0	25,259	0	23,961	79%	2161	8279	16,979	21,799	79%	3835	6435	18,823	20,127
**CASE F**	32%	0	25,259	0	23,961	77%	1976	9248	16,010	21,986	80%	3030	6628	18,630	20,932

**Table 4 sensors-21-02707-t004:** Accuracy metrics for Support Vector Machines (SVM) with linear kernel, polynomial kernel, and radial kernel. F1 score (F1), FN, FP, TN, and TP.

	SVM LINEAR					SVM POLYNOMIAL					SVM RADIAL				
	**F1**	**FN**	**FP**	**TN**	**TP**	**F1**	**FN**	**FP**	**TN**	**TP**	**F1**	**FN**	**FP**	**TN**	**TP**
**CASE A**	70%	5107	9286	9601	25,226	70%	3664	10,268	8621	26,667	70%	3578	10,558	8330	26,754
**CASE B**	70%	5041	9315	9572	25,292	70%	3677	10,172	8717	26,654	70%	3597	10,535	8353	26,735
**CASE C**	70%	4831	9407	9480	25,502	71%	3683	10,059	8829	26,649	70%	3622	10,495	8394	26,709
**CASE D**	77%	2030	9035	16,223	21,932	79%	712	9495	15,765	23,248	78%	650	9714	15,545	23,311
**CASE E**	77%	2033	9022	16,236	21,929	79%	717	9477	15,782	23,244	78%	650	9716	15,544	23,310
**CASE F**	77%	2032	9020	16,238	21,930	79%	717	9465	15,794	23,244	78%	649	9717	15,543	23,311

## Data Availability

Not applicable.

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
