# Peer review of "Towards Dependable IoT via Interface Selection: Predicting Packet Delivery at the End Node in LoRaWAN Networks"

_sensors, 2021, doi:10.3390/s21082707_

Round 1

Reviewer 1 Report

Very nice work. I have a few quiestions.

1.- With the long analysis and a lot of measurement made. Is possible to obtain a formula to prevent the PLR.

2.- Distance of modules are very short and LoRa is a network that permit distance up to 10 km. Why not put the nodes at more longer distances?

Author Response

Dear reviewer, find attached a detailed response to the posted comments.

Thank you very much for the time taken to perform the review.

Reviewer 2 Report

This paper is interesting and well written. The availability of the developed tools on github is also highly appreciated.

I have only minor comments:

  • The meaning of the acronym CCA is never explicitly reported, neither the way this functionality works.
  • Figures 4 and 5 do not appear in the printed version of the paper, although they are visible in the electronic file.

Author Response

(The authors gave the same response as above.)

Reviewer 3 Report

The paper addresses an interesting subject and the results obtained can be claimed by the authors.

Citing more relevant research papers would better place the paper in the appropriate field. The reference list can be supplemented with more scientific papers from journals, conferences or books.

The authors could include a status diagram for the software application that is responsible for prediction for a better understanding of the operations performed in the case of a proposed model. Although the purpose of the paper is not to present PCA, LDA or SVM, it would be interesting for the readers to be able to get a clearer idea of the complexity of the prediction application.

I did not find any information about the software tools or instruments used for measuring certain communication parameters. Are such tools required or one can rely on the information provided by chips like SX1301 and SX1276?

I think more details about the hardware used for testing would be interesting and useful for the readers. The test scenario in Figure 1 is too briefly presented.

Would it be possible to complete the paper with information regarding the energy consumption? This is an important issue when talking about wireless nodes. How much extra wakeup time would be required for a wireless node if it would run different models in comparison to the base model?

What would be the computation time on the system you did the experiments on? Is it a PC or a microcontroller system, because then this information would be very useful?

The pdf document has a major problem when it is opened, the file is corrupted or it was generated incorrectly. After opening, Adobe Reader used about 1 GB of memory and is not responding each time when the cursor is moved.

Author Response

(The authors gave the same response as above.)
